# A grass-specific cellulose–xylan interaction dominates in sorghum secondary cell walls

Yu Gao [1,2], Andrew S. Lipton [3], Yuuki Wittmer[4], Dylan T. Murray [4] & Jenny C. Mortimer [1,2,5]✉

Sorghum (*Sorghum bicolor* L. Moench) is a promising source of lignocellulosic biomass for the production of renewable fuels and chemicals, as well as for forage. Understanding secondary cell wall architecture is key to understanding recalcitrance i.e. identifying features which prevent the efficient conversion of complex biomass to simple carbon units. Here, we use multi-dimensional magic angle spinning solid-state NMR to characterize the sorghum secondary cell wall. We show that xylan is mainly in a three-fold screw conformation due to dense arabinosyl substitutions, with close proximity to cellulose. We also show that sorghum secondary cell walls present a high ratio of amorphous to crystalline cellulose as compared to dicots. We propose a model of sorghum cell wall architecture which is dominated by interactions between three-fold screw xylan and amorphous cellulose. This work will aid the design of low-recalcitrance biomass crops, a requirement for a sustainable bioeconomy.

[1] Joint BioEnergy Institute, Emeryville, CA 94608, USA. [2] Environmental Genomics and Systems Biology Division, Lawrence Berkeley National Laboratory, 1 Cyclotron Road, Berkeley, CA 94720, USA. [3] Environmental Molecular Sciences Laboratory, Pacific Northwest National Laboratory, Richland, WA 99354, USA. [4] Department of Chemistry, University of California, Davis, CA 95616, USA. [5] School of Agriculture, Food and Wine, Waite Research Institute, Waite Research Precinct, University of Adelaide, Glen Osmond, SA 5064, Australia. ✉email: jcmortimer@lbl.gov

Lignocellulosic biomass, a renewable source of organic carbon, will be a major feedstock for sustainable biorefineries producing bio-based fuels, chemicals, and materials[1,2]. These biorefineries are required to address the twin global demands of increased energy availability[3,4] and reduced emissions[5]. Grasses, such as sorghum (*Sorghum bicolor* L. Moench), *Miscanthus × giganteus*, and switchgrass (*Panicum virgatum* L.), are promising as dedicated bioenergy crops, due to their high photosynthetic efficiency and ability to grow in a range of environmental conditions[4,6]. However, plant cell walls, which comprise most of lignocellulosic biomass, have evolved to provide many crucial functions to the plant, including structural support, and protection from herbivory and pathogens[7]. The resulting highly recalcitrant cell wall architecture is an impediment to the efficient deconstruction of plant cell walls into simple monomers[8,9]. Improved understanding of the architecture of intact grass plant cell walls at a molecular level will provide key insights into this recalcitrance, and support the predictable design of both the biomass deconstruction process, as well as the development of engineered bioenergy crops.

All plant cells are surrounded by a thin, extensible, primary cell wall. Thickened secondary cell walls are only deposited in certain cell types at the cessation of cell expansion, but form the majority of biomass. Secondary cell walls are composed of polysaccharides (e.g., cellulose, xylan) and a complex polyphenolic network of lignin. Cellulose makes up 35–45% of the dry weight of grass secondary cell walls[10]. It consists of flat-ribbons of α-1,4-linked glucan chains which self-assemble into a crystalline fibrillar structure via hydrogen bonding. The resulting fibril has surfaces with distinct hydrophilicity that allow potential interactions with other cell wall components. However, cellulose crystallites are not perfect, and there is a significant fraction of cellulose that is less ordered, referred to as amorphous cellulose. Hence, two major domains of cellulose can be distinguished: a crystalline cellulose domain and an amorphous cellulose domain. Xylan is the major hemicellulose, which makes up 20–35% of the dry weight of grass secondary cell walls[11–13]. Xylan has a α-1,4-linked xylosyl unit backbone. However, unlike in dicot plants, grass xylan is highly substituted by α-L-arabinofuranosyl units at the *O*-2 and *O*-3 positions on the xylosyl units of the backbone and comparatively low density of substitution with α-1,2-glucuronosyl or α-1,2-(4-*O*-methyl)-glucuronosyl units[14]. This xylan structure is known as glucuronoarabinoxylan (GAX), though the exact substitution pattern is dependent on tissue and species. In addition, grass xylan is also substituted with acetyl groups on the backbone xylosyl units at the *O*-2 and *O*-3 positions, and some arabinosyl units can also carry acetyls at the *O*-2 position[13,15]. Grass xylan is acetylated to a lesser extent compared to dicot plants[15]. Xylans are believed to be present in two major conformations, two-/three-fold screw conformations, and the conformation is highly dependent on the pattern of substitution. For instance, the even pattern of acetyls and glucuronosyl/4-*O*-methyl-glucuronosyl units on *Arabidopsis thaliana* (Arabidopsis) allows these substitutions to lie on one side of the xylan backbone, where every two xylosyl units have a 360° turn (two-fold screw) forming a flat-ribbon shape. Without this even pattern, the xylan will have a 360° turn every three xylosyl units (three-fold screw) forming a helical shape due to steric hindrance[16]. Additionally, another characteristic feature of grass secondary cell wall is the presence of hydroxycinnamic acids (i.e., ferulic acid and *p*-coumaric acid), which are esterified to the *O*-5 of the α-1,3-arabinosyl substitutions on grass xylan backbone and can be esterified and/or etherified to lignin[10,11,13,17,18]. Ferulate units on grass xylan facilitate the covalent cross-link of xylan-xylan and xylan-lignin by formation of diferulates through dehydrodimerization and participation in lignification via radical coupling to bridge the lignin and polysaccharides, and stiffen the cell wall and maintain structural integrity[11,18–23].

Although the chemical structures of individual cell wall components are relatively well defined, the interaction between the cell wall components that form the three-dimensional network remains poorly understood for grasses. Methods to access the molecular information of intact plant cell walls were limited by the need for additional chemical extraction and pre-treatments, until a groundbreaking effort by Hong and colleagues which employed multi-dimensional solid-state Nuclear Magnetic Resonance (ssNMR) to reveal the architecture of native primary cell walls in Arabidopsis[24–29]. More recently, the architecture of secondary plant cell walls was also investigated via ssNMR. Simmons et al. found direct evidence for the presence of xylan in both two- and three-fold screw conformations in the dicot secondary cell wall. In a cross-polarizing experiment, which emphasizes the immobile components, the xylan-cellulose interaction was dominated by xylan in a two-fold screw conformation, with only a minor proportion of three-fold xylan detected[30]. Terrett et al. reported similar xylan-cellulose interactions in the secondary plant cell walls of softwoods due to the even pattern of substitution of softwood GAX. Additionally, galactoglucomannan, the other major hemicellulose in softwoods, binds the same cellulose microfibrils. Both cellulose-bound xylan and galactoglucomannan are also associated with lignin[31]. Kang et al. employed ssNMR to investigate the lignin-polysaccharide interactions in the secondary plant cell walls of maize (*Zea mays*). Results showed a conformation-dependent bridge behavior of xylan in the secondary plant cell walls, linking the cellulose fibrils and lignin, in which two-fold screw xylan coats the cellulose microfibril hydrophilic surface and the three-fold screw xylan connects the lignin nanodomain via electrostatic interactions[32].

Using Arabidopsis with genetically engineered xylan structures, Grantham et al. determined that an even pattern of xylan substitution is critical for the formation of the two-fold screw conformation and its interaction with cellulose[16]. Evenly patterned decorations on the xylan backbone allow the substitutions to orient along one side of the molecule, leaving the undecorated side of the resultant flat-ribbon shape free to hydrogen bond with the hydrophilic surface of the cellulose microfibril[16,33,34]. Although the substitution pattern of the xylan backbone in grasses is not known, the large amounts of arabinosyl units on grass xylan suggests that it may not follow the dicot and softwood pattern, which may in turn alter its interaction with cellulose[11,12,17].

Here, to explore xylan-cellulose interactions in grasses, we perform multi-dimensional ssNMR analysis on sorghum to reveal the native architecture of its secondary cell wall. The results suggest a lack of two-fold screw xylan, with the majority of the xylan showing a three-fold screw conformation. However, this three-fold screw xylan shows relatively high rigidity as compared to Arabidopsis and softwood reported earlier, and close proximity with less ordered amorphous cellulose. We propose that sorghum xylan-cellulose interactions are dominated by xylan in the three-fold screw conformation and amorphous cellulose, in contrast to the interactions between xylan in the two-fold screw conformation and crystalline cellulose in softwoods and dicot plants. We also show that the fraction of amorphous cellulose in the sorghum secondary cell wall is approximately three-fold higher than that in Arabidopsis, a model dicot plant. These findings provide molecular level understanding of the grass cell wall structure. Accurate cell wall models will enable a predictive understanding of biomass deconstruction by identifying the most recalcitrant aspects of the architecture. These models will also aid the identification of molecular targets for developing bioenergy crops with improved biomass properties.

## Results

**Generation of [13]C-enriched sorghum tissue.** Sorghum plants were grown hydroponically in a $^{13}CO_2$-containing atmosphere in our in-house [13]C-growth chamber, as previously described[35]. Upon harvest, all tissue was immediately frozen in liquid nitrogen, and sliced into 1–2 mm pieces to enable packing into the rotor, preventing them from thawing. No further processing of the tissue was performed, to allow us to investigate a cell wall architecture as close to native as possible. Stem internode tissue (dominated by secondary cell walls) from the 3rd internode was divided into upper, middle, and lower (since it represents a developmental series, with the most mature at the base). Leaf and root tissue (dominated by primary cell walls) was also retained and used for comparison.

**Sorghum [13]C enrichment and cell wall composition.** A sample of the harvested tissue was processed into alcohol insoluble residue (AIR), and used for compositional analysis. [13]C enrichment was confirmed by following the procedure described in[35], and all tissue used in this study had a [13]C incorporation rate of over 90% (Supplementary Fig. 1). Monosaccharide composition of the non-cellulosic fractions of the cell wall were determined by HPAEC-PAD analysis after mild acid hydrolysis, and were comparable to previously reported data (Supplementary Table 1).

**ssNMR of sorghum stem internodes.** To characterize different parts of the same internode, quantitative direct polarization (DP) one-dimensional (1D) [13]C NMR experiments, which detects all the [13]C carbons present in the sample, were performed with a long (30 s) recycle delay (Supplementary Fig. 2). No major difference was detected between the upper, middle, and lower sections of the same internode. From this point on, stem data were collected only on the lower part of the internode.

To deconvolute the overlapping signals from the 1D spectra (Supplementary Fig. 2) and obtain an overview of the chemical structure of native secondary cell wall components, the lower internode sample was investigated by two-dimensional (2D) double-quantum (DQ) and single-quantum (SQ) [13]C-[13]C correlation experiments using the refocused INADEQUATE sequence with cross polarization (CP)[36]. This measures the chemical shifts of directly bonded [13]C nuclei from relatively immobile components of the cell wall (Fig. 1). The DQ chemical shift represents the sum of the SQ chemical shifts of two directly bonded carbons. Chemical shifts were assigned with reference to previous reports[24,25,30–32] (Supplementary Table 2). Few signals from pectin and xyloglucan were detected in these experiments, due to both their relatively high mobility and low abundance in the secondary cell wall. Since the glucan chains in cellulose fibrils are polymorphic in structure, the two domains of cellulose were detected in at least three identified environments each[31,37]. The three distinguished environments for both amorphous and crystalline cellulose may arise from differences in hydrogen bonding patterns between glucan chains, slight changes in glucan chain conformation, variations in bond geometries, and changes and inconsistencies in neighboring chain environments within the microfibrils[38,39]. Here, for each carbon in a glucosyl unit from cellulose ($C_n$) we represent these different environments as follows: superscript [C] or [A] represents the crystalline or amorphous cellulose domain respectively, and superscript[1–3] represents the three distinguished environments of cellulose in each domain. The major chemical shift differences between the two domains of cellulose are present in the cellulose carbon-4 (C4) at ~89 ppm ([1–3C]C4) and ~84 ppm ([1–3A]C4), and C6 at ~65 ppm ([1–3C]C6) and ~62 ppm ([1–3A]C6). The signals from xylosyl units of three-fold screw xylan backbone were identified ([3f]Xn1–5 and [3f,A]Xn1–5, where superscript [2f] or [3f] represents the conformation of xylan and the superscript [A] represents the 3-O-arabinose

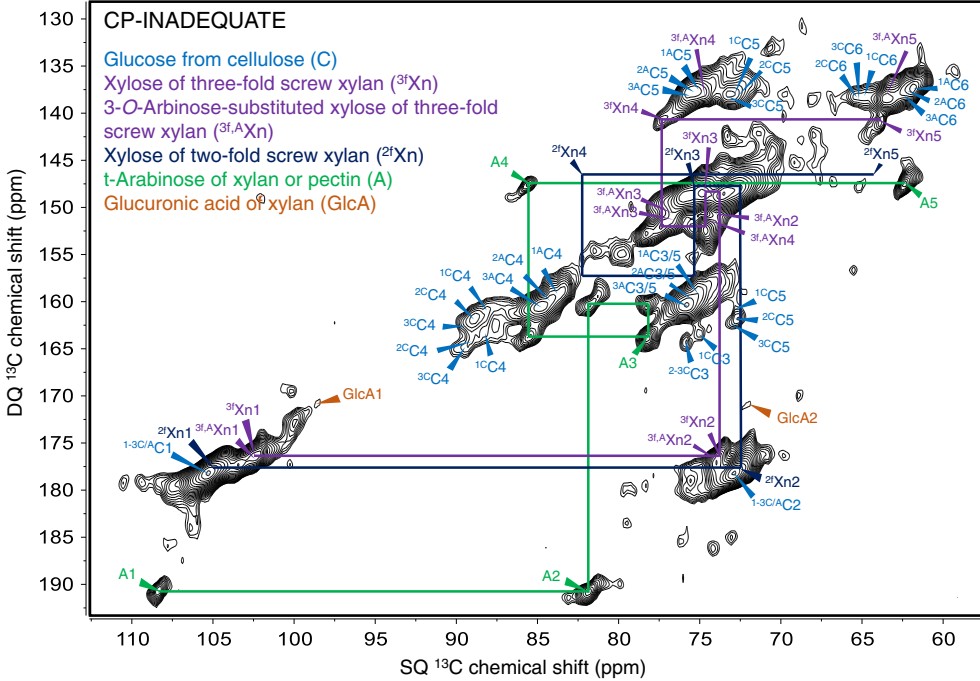

**Fig. 1 Immobile polysaccharides detected by refocused [13]C CP-INADEQUATE experiments in sorghum secondary cell walls.** The chemical shift assignments shown are derived from previously published chemical shifts (listed in Supplementary Table 2). Signals from pectin and xyloglucan are not detected due to their relatively high mobility, as well as their low abundance in the grass secondary cell wall. Glucosyl units (C1–C6) from cellulose in six distinguishable environments were assigned into two major domains, which are the crystalline [C] and amorphous [A] cellulose ([1–3C]C1–[1–3C]C6, [1–3A]C1–[1–3A]C6). A lack of signals from xylosyl units of two-fold screw xylan was observed. Instead, signals from immobile xylosyl units of rigid three-fold screw xylan were detected ([3f]Xn1–[3f]Xn5, [3f,A]Xn1–[3f,A]Xn5). High intensity signals from xylan arabinosyl units were also detected (A1–A5).

substituted xylosyl units of xylan), but there were a lack of signals from xylosyl units of two-fold screw xylan ($^{2f}$Xn1–5) (Fig. 1). In addition, high intensity signals from arabinosyl units decorating the xylan backbone (A1–A5; Fig. 1) were identified, which is consistent with our monosaccharide analysis on the non-cellulosic components from stem internodes (Supplementary Table 1). Lower intensity signals from carbons 1 and 2 of glucuronic acid were also detected (GlcA1 and GlcA2) due to the lower abundance of glucuronic acid substitutions on the xylan backbone, as compared to arabinose. A DP-INADEQUATE experiment with relatively short recycle delay, 2 s, was also performed on the sorghum stem internode samples to enhance the detection of the relatively mobile components in the secondary cell walls[25,30,32,40], and spectrum was labeled and shown in Supplementary Fig. 3.

Two-fold screw xylan with a flat-ribbon shape has been reported as the dominant conformation detected by CP in the secondary cell walls of dicot plants[30,34] and softwoods[31], which facilitates the binding with the cellulose fibrils on their hydrophilic surface via hydrogen bonding. However, our CP-INADEQUATE result indicated a significantly higher fraction of xylan was present in the three-fold screw conformation than in the two-fold screw conformation in the sorghum secondary cell wall. According to the intensities of cross peaks from the Xn4 and Xn5 of the xylosyl units from both two- (82.1 ppm, 146.4 ppm, and 64.3 ppm, 146.4 ppm) and three-fold (77.3 ppm, 141.1 ppm, and 63.8 ppm, 141.1 ppm) screw xylan backbone, the immobile three-fold screw xylan was the dominant conformation. This could be due to the large quantity of arabinosyl substitutions on the sorghum xylan backbone. Dupree et al. previously demonstrated that the spacing between the substitutions is critical for the formation of two-fold screw xylan due to the steric hindrance effect[16,30,33,34]. For instance, xylan in dicots has little or no arabinosyl substitution, and acetylations and glucuronic acid substitutions are evenly spaced on a major fraction of the xylan backbone, which allows the formation of two-fold screw conformation. Similarly, although softwood xylan is substituted by both glucuronic acid and arabinose, substitutions are evenly spaced on every six and two xylosyl units respectively on the xylan backbone. Hence, we speculate that a large number of closely spaced substitutions on sorghum xylan would likely disrupt any pattern of regular spacing on a major fraction of the xylan. The result of this would be more xylan in the three-fold screw conformation in sorghum secondary cell walls. This spacing pattern has yet to be confirmed, due to a lack of a glycosyl hydrolase with the required specificity for cleaving at an arabinosyl substitution, akin to the glucuronic acid-dependent xylanase XynC used to determine the spacing pattern in Arabidopsis[41]. However, since CP experiments emphasize the signals from relatively immobile components of the cell wall, the large amount of three-fold screw xylan detected suggests that this confirmation could be important for xylan-cellulose interactions in sorghum.

In contrast, in recent work by Kang et al., two-fold screw xylan was detected by CP-INADEQUATE as the dominant conformation present in maize, a grass closely related to sorghum. However, their CP-INADEQUATE data did not detect arabinose signals, which is surprising given the reported structure of maize xylan[17,42]. We hypothesized that this could be due to the additional lyophilization step they performed on the plant samples prior to analysis, which may alter the native wall structure. To test this, we lyophilized the sorghum stem tissue and rehydrated the sample according to the procedure that Kang et al. described in[32]. A CP-INADEQUATE experiment was conducted on the rehydrated sample and compared with our previously collected CP-INADEQUATE spectrum on the untreated sample (Supplementary Fig. 4). We found that the previously strong signals from arabinosyl units were lost in the spectrum of lyophilized-rehydrated sample (Supplementary Fig. 4).

The intensities of $^{3f}$Xn4 and $^{3f}$Xn5 also decreased in the lyophilized-rehydrated sample, suggesting that lyophilization may disrupt the immobile three-fold screw xylan interactions with other wall components. Rehydration of the sample did not restore these interactions, but instead, this free three-fold xylan became more mobile in the rehydrated water, and therefore no longer detectable by the CP. In addition, we also collected DP-INADEQUATE spectrum on the lyophilized-rehydrated sample and compared it with the previously collected DP-INADEQUATE spectrum from the untreated sample (Supplementary Fig. 5). The spectra show that there is a significant enhancement of signals from arabinosyl and xylosyl units of the three-fold screw xylan in the lyophilized-rehydrated sample, which indicates a significant amount of arabinosyl and xylosyl units from three-fold screw xylan have become more mobile after lyophilization and rehydration. On the other hand, the cross peak intensity from crystalline cellulose and two-fold screw xylan was enhanced in the lyophilized-rehydrated sample. Two-fold screw xylan has a flat-ribbon shape that can form a crystalline structure that is similar to cellulose. Lyophilization of the sample increases the crystallinity of these structures[43]. Although rehydration of the sample could reduce crystallinity to a certain extent, the cell wall architecture will remain altered permanently as compared to native cell walls. Hence, the increased rigidity of two-fold screw xylan and cellulose led to enhanced CP signals for the lyophilized-rehydrated sample.

For comparison, we also performed additional CP-INADEQUATE experiments on never-dried sorghum leaf and root samples of sorghum, which are richer in primary cell walls, for comparison with the stem internode samples. The leaf samples show that the majority of xylan detected by CP-INADEQUATE are in a three-fold screw conformation, but to a lower extent compared to the stem (Supplementary Fig. 6, top panel). However, analysis of the root material indicates that there is almost no immobile xylan in either conformation, as detected by CP-INADEQUATE (Supplementary Fig. 6, bottom panel).

**Three-fold screw xylan binds to amorphous cellulose.** Xylan in the intact plant cell wall can populate both mobile and immobile states[30–32,40]. The states can be distinguished based on the extent of their interactions with the immobile cellulose, which are mediated via either hydrogen bonding or Van der Waals forces: the cellulose-bound fraction of xylan is immobile, whereas the fraction of xylan that fills the inter-microfibril space is highly mobile[29–32,44–46]. In contrast to reported data for dicots, we found that the three-fold screw xylan, not two-fold screw xylan, was the dominant conformation of xylan in the sorghum secondary cell wall, in both immobile and mobile forms (Fig. 1). To investigate how this impacts xylan-cellulose interactions, $^{13}$C-$^{13}$C proton-driven spin diffusion (PDSD) experiments were performed with CP with three different mixing times (30, 100, and 1500 ms). Due to the CP transfer used in the experiment, these measurements report on the immobile fraction of xylan in the sample characterized by limited larger-scale molecular motions about an average structure with motional timescales on the order of μs to ms. This is in contrast to the mobile fraction of xylan, which is characterized by much faster molecular reorientations with motional timescales on the order of ns to μs. PDSD experiments provide information on carbons in close spatial proximity. The longer the mixing time, the longer the distance observed on the spectra.

The short-mixing time (30 ms) CP-PDSD experiment is dominated by intramolecular peaks, such as carbons from glucan chains with six identified allomorphs in cellulose microfibrils, xylosyl units in the xylan backbone in both two- and three-fold screw conformations, and arabinosyl units from xylan (Fig. 2).

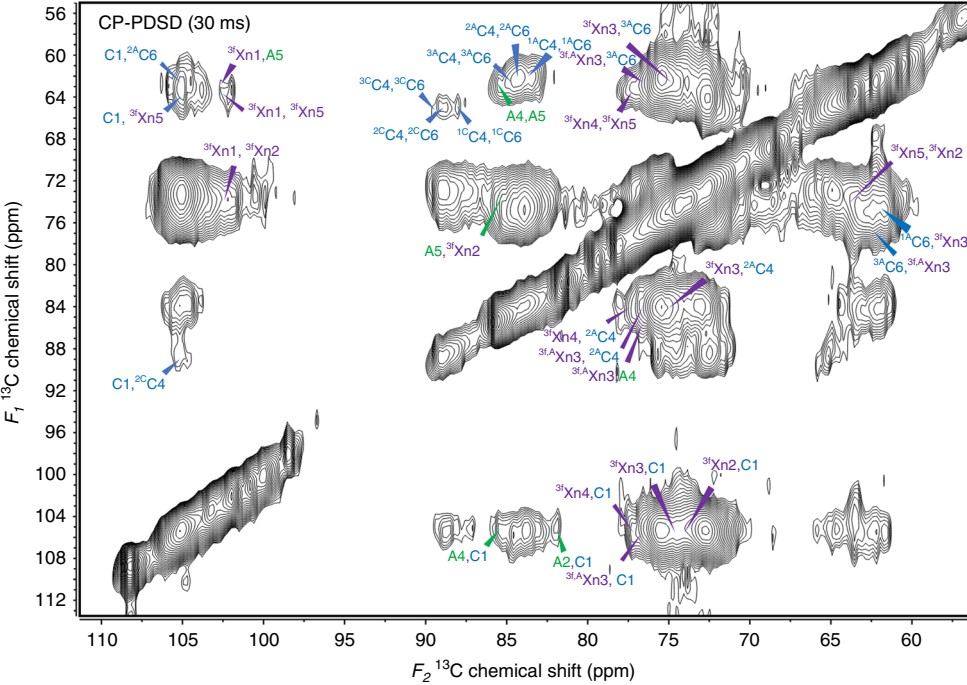

**Fig. 2 Close proximities between moieties from three-fold screw xylan and amorphous cellulose were detected in sorghum secondary cell walls by CP-PDSD experiments with a short-mixing time (30 ms).** Cross peaks indicate that the majority of intramolecular interactions and close intermolecular interactions between arabinosyl and xylosyl units from three-fold screw xylan and glucosyl units from amorphous cellulose were labeled in the spectrum.

In addition, some intermolecular cross peaks between cellulose and xylan are observed. There are no cross peaks representing the interaction between the two-fold screw xylan and either the crystalline or amorphous domain of cellulose, likely due to the limited amount of two-fold screw xylan in the sorghum secondary cell wall. Instead, cross peaks, such as $^{3f}Xn4$-$^{2A}C4$ (77.3 ppm, 89 ppm), $^{3f}Xn3$-$^{2A}C4$ (74.7 ppm, 89 ppm), $^{3f}Xn3$-$^{1A}C6$ (74.7 ppm, 61.8 ppm), $^{3f}Xn3$-$^{3A}C6$ (74.7 ppm, 62.5 ppm), $^{3f,A}Xn3$-$^{2A}C4$ (76.7 ppm, 84.1 ppm), and $^{3f,A}Xn3$-$^{3A}C6$ (76.7 ppm, 62.5 ppm), all indicate that the xylosyl units from xylan in a three-fold screw conformation is closely interacting with amorphous cellulose. Additionally, carbons from the xylosyl units of three-fold screw xylan, $^{3f}Xn2$ to $^{3f}Xn5$, show cross peaks with the C1 from cellulose (105.2 ppm), which derives from cellulose in multiple environments. However, since we observed no cross peaks between carbons from three-fold screw xylan and $^{C}C2$ to $^{C}C6$, we conclude that the $^{3f}Xn2$- to $^{3f}Xn5$-C1 cross peaks were primarily contributed by the C1 from amorphous cellulose. Furthermore, arabinosyl units from xylan show close interactions with the cellulose C1, as indicated by two cross peaks, A2-C1 (81.8 ppm, 105.2 ppm) and A4-C1 (85.6 ppm, 105.2 ppm). Since a low abundance of two-fold screw xylan was observed (Fig. 2), the arabinose signals are likely from three-fold screw xylan. The close interactions between the arabinosyl units and the cellulose imply that the less ordered amorphous cellulose is able to bind with helical three-fold screw xylan. In the CP-PDSD experiment with 100 ms mixing time, the spectrum showed similar intermolecular cross peaks to the 30 ms mixing time, but with enhanced intensities (Supplementary Fig. 7). Together, these data suggest the xylan-cellulose interaction is dominated by an immobile xylan with three-fold screw conformation and amorphous cellulose across short distances. It remains unclear to us what type of forces are facilitating such interactions, but we speculate it involves both Van der Waals contacts and some hydrogen bonds[16,47]. The less ordered amorphous cellulose may have a distorted flat-ribbon shape and therefore create more surface space to occasionally enable the formation of hydrogen bonds with the three-fold screw xylan on the hydrophilic side[48]. Interactions with Van der Waals forces are mainly from the hydrophobic surface of cellulose fibrils[16]. Hence, the xylan-cellulose interactions in sorghum secondary cell walls are significantly weaker than those in dicot plants and softwoods which are dominated by hydrogen bonds between two-fold screw xylan and cellulose fibrils on the hydrophilic surface.

To further explore the interaction between three-fold screw xylan and amorphous cellulose, we measured the spin-lattice relaxation times ($T_1$) at various chemical shifts representing different components (Fig. 3, Supplementary Table 3). The higher $T_1$ indicates slower molecular dynamics of the cell wall component. The results show that carbons from crystalline cellulose, such as C1, $^{1C}C6$, and $^{2C}C6$, have the highest $T_1$ values, ~9 s, and the carbons from amorphous cellulose, such as $^{1A}C3/5$, $^{1A}C6$, and $^{2A}C6$, have similar $T_1$ values as the carbons from arabinosyl (A2 and A4) and xylosyl ($^{3f}Xn2$ to $^{3f}Xn5$) units in relatively immobile three-fold screw xylan, ~5 s. $T_1$ measurements of the cell wall components further demonstrate that amorphous cellulose shares similar molecular dynamics with the relatively immobile fraction of the three-fold screw xylan, while crystalline cellulose has significantly reduced molecular motion. This supports our interpretation that amorphous cellulose and three-fold screw xylan are closely interacting with each other.

One-dimensional spectra were extracted at seven chemical shifts (108.4, 105.2, 102.5, 84.1, 81.8, 63.8, and 62.4 ppm) from the $F_1$ plane of the CP-PDSD spectra with both 30 and 1500 ms mixing times and compared in Supplementary Fig. 8. No interaction between the three-fold screw xylan and the crystalline cellulose was detected in the short-mixing time (Supplementary Fig. 8). This is consistent with previous work reported by Dupree et al. using Arabidopsis, which showed that the flat-ribbon shape of two-fold screw xylan with even pattern of substitutions is required for binding on the highly ordered crystalline cellulose hydrophilic surface[16,30,34]. In addition to the intramolecular interactions, many

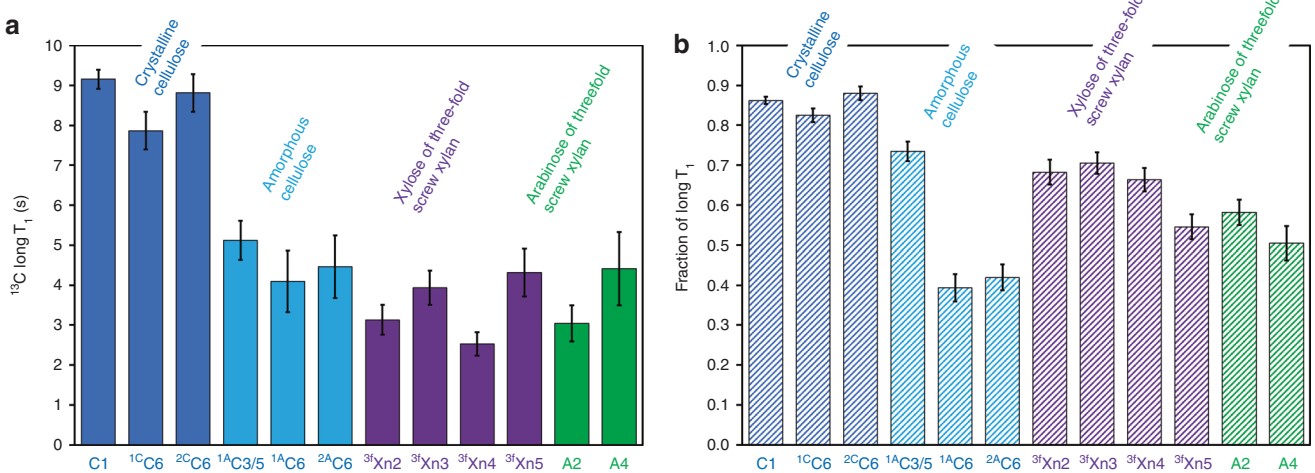

**Fig. 3 Spin-lattice relaxation time (T₁) measurements indicate immobile three-fold screw xylan has similar molecular dynamics to amorphous cellulose in sorghum secondary cell walls.** The long $T_1$ of moieties from both domains of cellulose and three-fold screw xylan (**a**) and the fraction of the long T1 component (**b**) are shown. Data are presented as mean values ± standard errors of the mean (SED). Error bars are SED of the fitting parameters on 20 data points. Fitted data are also provided in Supplementary Table 3.

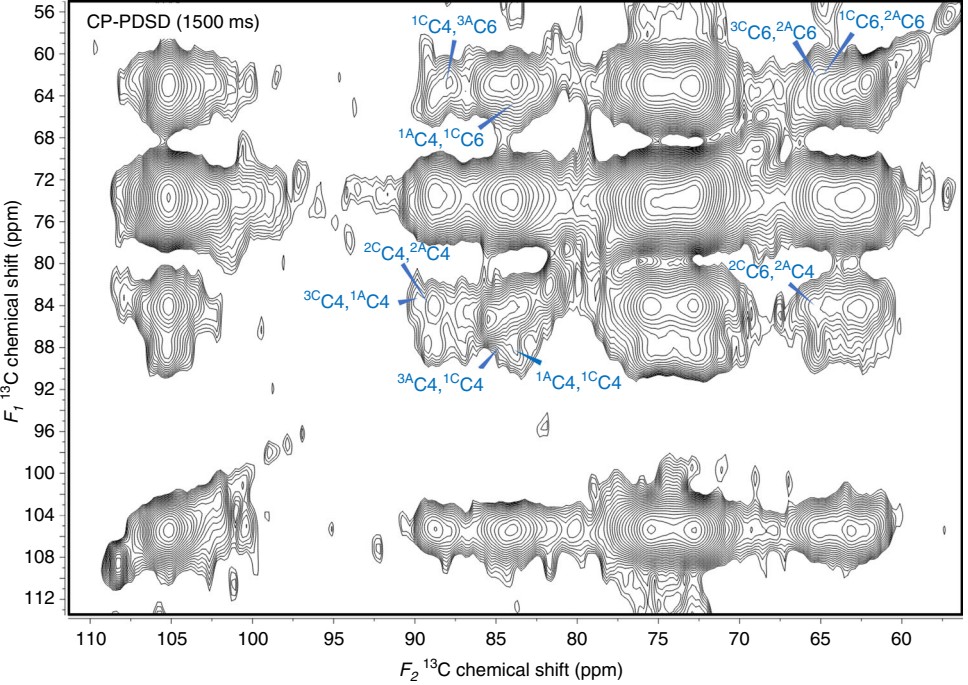

**Fig. 4 Interactions between crystalline and amorphous cellulose were detected in sorghum secondary cell walls by CP-PDSD experiments with a long-mixing time (1500 ms).** Cross peaks representing interactions between crystalline and amorphous cellulose are labeled.

more intermolecular interactions were also detected with long-mixing time (Supplementary Fig. 8). Although cellulose domains in different environments were expected to be close to each other, cross peaks between crystalline and amorphous cellulose were only detected in the long-mixing time (1500 ms) CP-PDSD experiment (Fig. 4 and Supplementary Fig. 8). Cross peaks between C4s (~89 ppm for crystalline cellulose, ~84 ppm for amorphous cellulose) and C6s (~65 ppm for crystalline cellulose, ~62 ppm for amorphous cellulose) from cellulose in different environments were observed (Fig. 4). We interpret this as being due to the relatively low ratio of crystalline to amorphous cellulose in the sorghum cell wall, as described in the following experiments, and that crystalline cellulose is spatially further away from the amorphous cellulose than the three-fold screw xylan.

**A high fraction of cellulose in sorghum is amorphous.** Since the sorghum secondary cell wall is dominated by three-fold screw xylan and amorphous cellulose interactions, we next investigated how sorghum cell wall architecture, in particular the xylan-cellulose interactions, compare to a wall architecture that is dominated by interactions between two-fold screw xylan and highly ordered crystalline cellulose, such as that described for dicots and softwoods[30,31]. To enable a direct comparison of sorghum with Arabidopsis (which is the best characterized dicot secondary cell wall), we also produced [13]C-enriched Arabidopsis stems and analyzed them using CP-PDSD with a short-mixing time (30 ms). Figure 5 shows the overlay of the resulting spectra in four regions that illustrate the intramolecular cross peaks within cellulose. The intensities were normalized by the cross

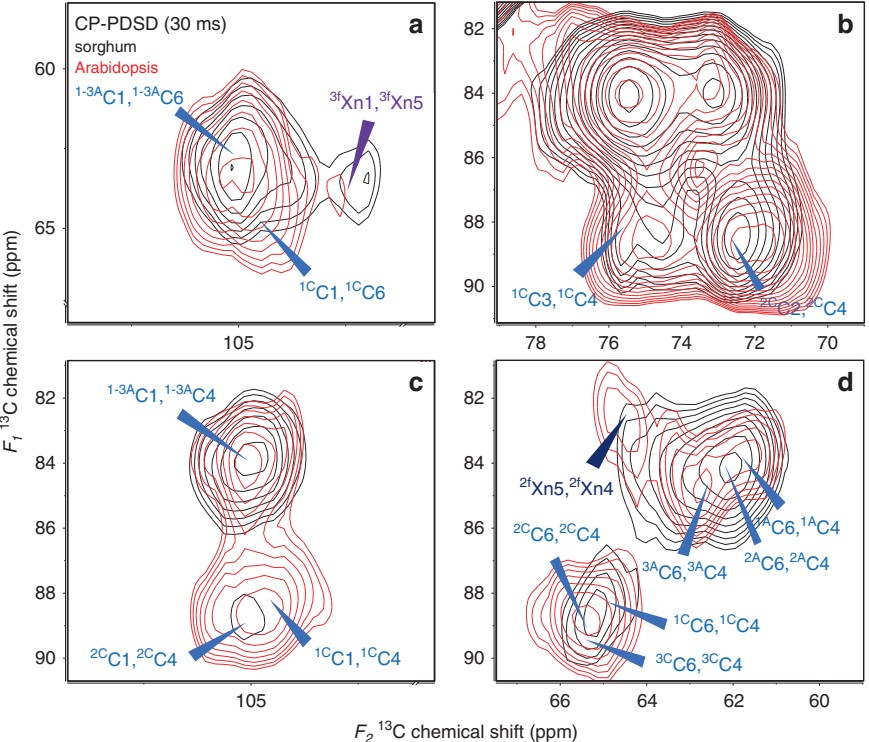

**Fig. 5 Sorghum secondary cell walls have a lower ratio of crystalline to amorphous cellulose than Arabidopsis.** An overlaid CP-PDSD experiment with short-mixing time (30 ms) on sorghum and Arabidopsis stem tissue shows intramolecular interaction cross peaks of cellulose in different environments in four regions. **a** Interactions between C1 and C6 of cellulose. **b** Interactions between C2 and C4, and C3 and C4 of cellulose. **c** Interactions between C1 and C4 of cellulose. **d** Interactions between C4 and C6 of cellulose. Sorghum shows significantly higher intensity of immobile three-fold screw xylan cross peaks in panel (**a**), but much less intensity of two-fold screw xylan cross peaks in panel (**d**) than Arabidopsis. All four regions demonstrated fewer crystalline cellulose intramolecular cross peaks were detected in sorghum than in Arabidopsis.

peaks from intramolecular interactions between $^{1-3A}C1$–$^{1-3A}C6$ (~105 ppm, ~62 ppm) of amorphous cellulose (Fig. 5a). In the sorghum spectra, the high intensity cross peak of $^{3f}Xn1$–$^{3f}Xn5$ (102.5 ppm, 63.8 ppm) in Fig. 5a and the relatively low intensity of cross peak of $^{2f}Xn4$–$^{2f}Xn5$ (82.2 ppm, 64.3 ppm) in Fig. 5d, as compared to the Arabidopsis, indicates that the sorghum stem secondary cell wall is dominated by three-fold screw xylan. This is consistent with our CP-INADEQUATE results (Fig. 1).

Additionally, Fig. 5a–d shows that cross peaks from the intramolecular interactions of crystalline cellulose, such as $^{1C}C1$–$^{1C}C6$ (104.5 ppm, ~64.8 ppm), $^{1C}C3$–$^{1C}C4$ (75.8 ppm, 88.1 ppm), $^{2C}C2$–$^{2C}C4$ (72.8 ppm, 89 ppm), $^{1-2C}C1$–$^{1-2C}C4$ (~105 ppm, ~89 ppm), and $^{1-3C}C6$–$^{1-3C}C4$ (~65 ppm, ~89 ppm), were significantly diminished in sorghum compared to Arabidopsis. This is in contrast to the intensities of cross peaks from the intramolecular interactions of amorphous cellulose, which are similar between sorghum and Arabidopsis. To obtain the ratio of crystalline and amorphous cellulose, integrations were performed on cross peaks from the intramolecular interactions of all crystalline cellulose between C6 and C4 (~65 ppm, ~89 ppm) and all amorphous cellulose (~62 ppm, ~84 ppm). The ratio determined for Arabidopsis is 0.8:1, which is consistent with previous studies[31,49]. However, sorghum has a crystalline to amorphous cellulose ratio of 1:3, which is approximately three folds more amorphous cellulose in sorghum than in Arabidopsis and softwoods[31,49]. This is also verified by the relatively more quantitative 1D $^{13}C$ CP experiments, from which the crystalline and amorphous cellulose ratio were determined based on the integration of deconvoluted crystalline and amorphous cellulose C4 peaks. Similar results to the cross peak analysis were obtained from the 1D experiments (Supplementary Fig. S9). As a

corresponding phenomenon to the low ratio of two- and three-fold screw xylan, more less-ordered amorphous cellulose is found in the sorghum secondary cell walls, which would facilitate interactions with the three-fold screw xylan.

## Discussion

Recent studies of plant cell wall architecture using multi-dimensional ssNMR techniques have meant that the molecular details of plant cell walls are better understood than ever for various plant species[24–26,28–32,40,44,46]. However, there is a tremendous amount of information still to be elucidated regarding the networks and interactions between cell wall components. This is especially true for the grasses, since digestibility/recalcitrance of the cell wall is a key feature determining their application in bioenergy, animal feed, and human health[50–53]. Previous work demonstrated that the presence of two-fold screw xylan, which requires an even pattern of substitution on its backbone, is critical for interactions with the cellulose microfibrils on the hydrophilic surface in plant cell walls[16,30–32,34]. However, how these xylan-cellulose interactions are formed when such two-fold, evenly substituted xylan is present in low abundance (such as in grass secondary cell walls), remains unclear.

We found a lack of interactions between two-fold screw xylan and cellulose, and instead observed a close proximity between three-fold screw xylan and cellulose in the short-mixing time (30 ms) CP-PDSD spectrum. Additionally, the cellulose that shows closest proximity to the three-fold screw xylan is mostly the less ordered amorphous cellulose. Although the interaction with highly ordered crystalline cellulose on hydrophilic surface via hydrogen bonds requires xylan to be also oriented in a flat-ribbon morphology, the high fraction of helical three-fold screw xylan

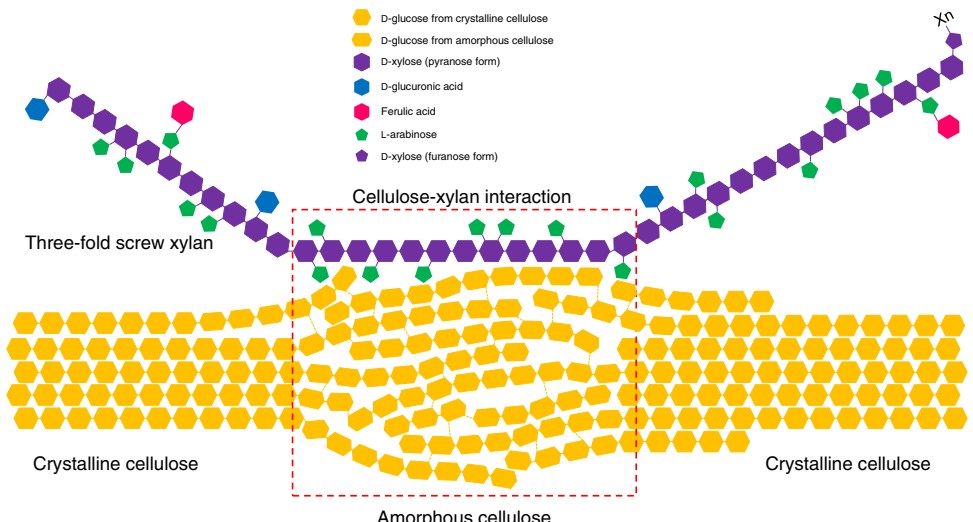

**Fig. 6 Model of xylan-cellulose interaction in sorghum secondary cell walls.** Frequent and irregular arabinosyl substitution on the xylan backbone reduces the fraction of xylan in a two-fold screw conformation, and the majority of xylan is in a three-fold screw conformation. Three-fold screw xylan rarely docks on the hydrophilic surface of the highly ordered, flat-ribbon shape crystalline cellulose due to its helical shape. Instead, it is found in association with less ordered amorphous cellulose, likely via Van der Waal forces and limited hydrogen bonds. A large fraction of cellulose in the sorghum secondary cell walls is amorphous cellulose, which may facilitate the interactions between xylan and cellulose.

may promote relatively weak interactions with the less ordered amorphous cellulose domain (Fig. 6) via Van der Waals contacts and occasional hydrogen bonds. Unlike the secondary cell walls of dicots and softwoods, due to the low fraction of two-fold screw xylan, grass xylan-cellulose interactions are dominated by relatively weak interactions between the amorphous cellulose and three-fold screw xylan. Hence, covalent links between xylan and lignin, via the dihydoroxycinammic acid linkages likely contribute more to cell wall strength in grasses. The cell wall strength/connectivity in grasses, important for agronomic traits such as lodging, can likely be enhanced via ferulic-acid mediated xylan-lignin interactions, and can be tested in future sorghum engineering efforts. Since the xylan-lignin interactions greatly contribute to cell wall recalcitrance for enzymatic degradation, the ability to tune the cross-links between xylan and lignin, for example in specific cell types e.g. leaves, may be an important strategy for tailoring sorghum for more efficient bioenergy production[19].

Li et al. have demonstrated that increasing the degree of arabinosyl substitution in GAX from *Miscanthus* (a grass) enhanced the biomass enzymatic digestibility[54]. The researchers also found that GAX-derived arabinose was released by cellulase digestion that preferentially took place at the amorphous region, but not by xylanase, which suggested that the xylan which is densely substituted with arabinosyl units is closely associated with amorphous cellulose, consistent with our findings. Similar effects were later reported in other grasses, including rice[55,56], sweet sorghum[57], wheat[56], and switchgrass[58,59]. Our findings provide direct evidence for interactions between three-fold screw xylan and amorphous cellulose, which uncovers the functional role of highly substituted xylan in grass in sorghum secondary cell wall architecture. This work provides foundational knowledge to guide future genetic engineering approaches for the design of cell walls in improved bioenergy, forage and food crops.

## Methods

**¹³C Plant materials**. Sorghum seeds (Tx430), a kind gift from Professor Jeff Dahlberg, UCANR, were sown on soil (PRO-MIX 'HP' with Mycorise PRO) and germinated in a controlled environment room (30 °C, 30% humidity, 150 µmol m$^{-2}$ s$^{-1}$, 14 h light/10 h dark) for 17 days with four leaves. Seedlings were then transferred into a 12 L plastic container (purchased from U. S. Plastic Corp.) containing 11 L of

Hoagland's nutrient solution pH 6.0, and placed in a self-constructed ¹³C enrichment growth chamber (28 °C, 70% humidity, 155 µmol m$^{-2}$ s$^{-1}$, 18 h light) for another 29 days, as previously described[35]. Stem tissue was harvested from the three internodes, and then split into three: "upper", "middle" and "lower". Root and leaf tissue were also harvested at the same time.

Arabidopsis seeds (ecotype Columbia-0) were surface sterilized and sown on half Murashige and Skoog salts including vitamins (Sigma, M5524-1L), agar (0.7% w/v) and sucrose (1% w/v). Following stratification (48 h at 4 °C in the dark), the plates were transferred into a growth chamber (23 °C, 150 µmol m$^{-2}$ s$^{-1}$, 15 h light/9 h dark) and grown vertically for 10 days. Seedlings were transferred into a 5 L plastic container with 4 L of Arabidopsis hydroponic solution, as described previously[60] and placed into the ¹³C enrichment chamber (24 °C, 60% humidity, 155 µmol m$^{-2}$ s$^{-1}$, 18 h light) for 29 days.

All harvested ¹³C-plant materials were snap frozen in liquid nitrogen, and stored at −80 °C until required.

Lyophilized-rehydrated sorghum sample were prepared by lyophilization of ¹³C-enriched sorghum stem internode tissue (~30 mg) overnight. Lyophilized samples were then rehydrated by adding 40 wt% deionized water and sonicated for 30 min before packing into a rotor for solid-state NMR analysis[32].

**Determination of ¹³C incorporation**. AIR was prepared from the ¹³C-enriched plant materials[61]. Sorghum AIR (1 mg) was pretreated with 4 M NaOH (50 µL) for 1 h at room temperature, then neutralized with 1 M HCl. Ammonium acetate buffer (pH 5.5, final concentration of 100 mM, final volume of 500 µL) was added to the sample, followed by digestion with endo-1,4-α-xylanase (Megazyme, rumen microorganism, GH11) overnight at room temperature. Arabidopsis AIR (1 mg) was incubated in ammonium formate (pH 5.5, 50 mM, 250 µL) with 4 U/mL xyloglucan-specific endo-α-1,4-glucanase (Megazyme, *Paenibacillus sp.*, GH5) for overnight at 37 °C.

The supernatants, containing released oligosaccharides, were mixed with 0.5% (w/v) of 2,5-dihydroxybenzoic acid in 50% methanol in water with 0.1% (v/v) TFA, and spotted on an MTP 384 target plate (Bruker). Matrix-assisted laser desorption/ionization-time-of-flight (MALDI-TOF) mass spectrometry (UltrafleXtreme instrument, Bruker) analysis was used to determine the mass and relative abundance of the released oligosaccharides. The oligosaccharides XAXX[62] and LXXG/GXXXG[63], were used to determine the ¹³C incorporation into the sorghum and Arabidopsis tissue, respectively. The ¹³C incorporation of sorghum and Arabidopsis used in this study was 92 and 91%, respectively.

**Magic angle spinning solid-state NMR**. For the sorghum stem internodes samples and Arabidopsis stem sample, solid-state ¹³C MAS NMR experiments were performed at the Environmental Molecular Sciences Laboratory (EMSL) on an Agilent 850 WB with a direct drive (VNMRS) NMR console. The never-dried ¹³C-enriched tissue was sliced into ~1–2 mm pieces using a razor blade on a cold surface (−20 °C). Samples (~30 mg) were packed into 3.2 mm zirconia rotors with O-ring seals (Revolution NMR). All ¹³C NMR spectra were obtained at 20 °C and with a static magnetic field strength of 20.0 T (Larmor frequencies of 849.727 MHz and 213.685 MHz for ¹H and ¹³C irradiation, respectively). The NMR probe

utilized was an Agilent 3.2 mm HXY MAS probe tuned in double resonance mode, and the MAS frequency was 13 kHz. CP MAS acquisition was accomplished with a standard ramped CP pulse sequence using a 4 μs $^1$H 90° pulse, a 1 ms contact pulse (ramped $^1$H RF amplitude[64]), and a 2 s recycle delay. SPINAL-64 decoupling[65] was applied during acquisition at a $^1$H nutation frequency of 62–80 kHz. $^{13}$C chemical shifts are reported relative to tetramethylsilane [$(CH_3)_4Si$] by assignment of the methylene peak in adamantane at 38.48 ppm as an external secondary standard. Two-Dimensional (2D) DQ to SQ correlation experiments (through-bond) were performed via a refocused INADEQUATE[66,67] pulse sequence with either CP or DP. Through-space interactions were measured with SQ–SQ 2D PDSD[46,68] experiments with mixing times of 30 ms, 100 ms or 1.5 s. $T_1$ measurements were performed with a CP analogue of an Inversion Recovery experiment where the polarization transfer step is immediately followed by a π/2 $^{13}$C pulse to invert the carbon signal. The magnetization is then allowed to relax by a varying delay and then read out with another π/2 pulse. Fitting the signal versus the variable delay to a biexponential (see equation in Supplementary Table 3) allows the extraction of $T_1$ constants. All ssNMR data on sorghum stem internodes samples and Arabidopsis stem sample were collected by using VNMRJ software (v4.2 A) and analyzed by using MestReNova software (v14.0). $T_1$ measurement data were fitted by using OriginPro 2016 software.

For the sorghum leaf and root samples, solid-state NMR measurements were also recorded on a Bruker Avance wide-bore 11.7 T NMR spectrometer using a 4 mm triple resonance solenoid 4TR Bruker MAS probe. Samples of leaves and roots were removed from −80 °C storage and kept on dry ice until prior to packing into a 4 mm zirconia rotor and sealed with cyanoacrylate gel and a vespel cap. Experiments were recorded immediately after packing the rotor with the sample. The refocused CP-INADEQUATE spectra were obtained at room temperature with a spinning speed of 10 kHz, a $^1$H 90° pulse of 3.5 μs, a $^{13}$C 90° pulse of 4 μs, $^1$H-$^{13}$C CP with irradiation at ~70 kHz and 50 kHz on the $^1$H and $^{13}$C channels, respectively, for 1.0 ms with a $^1$H ramp of 20%, and 70 kHz TPPM $^1$H decoupling. The $t_1$ time increment was 20.0 μs and 250 States-TPPI points were collected for a 2.5 ms $t_1$ evolution time. The spin echo delay time was 4.2 ms. 128 scans were averaged with a recycle delay of 1.75 s for a total experiment time of 15.6 h. All ssNMR data on sorghum root and leaf samples were collected by using Bruker Topspin software (v2.1) and analyzed by using MestReNova software (v14.0).

**Reporting summary**. Further information on research design is available in the Nature Research Reporting Summary linked to this article.

## Data availability
All relevant data supporting the findings of this manuscript is available within the paper and its supplementary information files. All the unprocessed NMR data files are available at https://doi.org/10.25582/DATA.2020-09.2061884/1661467. Source data are provided with this paper.

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

## Acknowledgements

The authors thank Prof. Ray Dupree, University of Warwick, for helpful discussions. We thank Dr. Nancy M. Washton, Environmental Molecular Sciences Laboratory, for help with scheduling and set-up of the ssNMR experiments. This work was conducted as part of the DOE Joint BioEnergy Institute (http://www.jbei.org) supported by the U. S. Department of Energy, Office of Science, Office of Biological and Environmental Research, through contract DE-AC02-05CH11231 between Lawrence Berkeley National Laboratory and the U. S. Department of Energy. Part of this work was conducted by the Environmental Molecular Sciences Laboratory (grid.436923.9), a DOE Office of Science scientific user facility sponsored by the Department of Energy's Office of Biological and Environmental Research and located at PNNL under contract DE-AC05-76RL01830. The work was also supported by startup funding from the University of California, Davis.

## Author contributions

Y.G. and J.C.M. constructed the growth chamber and grew ¹³C plants. A.S.L., D.T.M., and Y.W. conducted the NMR experiments. Y.G., A.S.L., D.T.M, Y.W., and J.C.M. analyzed the experimental data. Y.G., J.C.M., A.S.L., and D.T.M wrote the manuscript.

## Competing interests

The authors declare no competing interests
