## [Peer Review File · Nature Communications]

REVIEWER COMMENTS

Reviewer #1 (Remarks to the Author):

The study by Gao et al. explores the interactions between xylene and cellulose in sorghum, in order to better understand its wall architecture and ultimately provide guidance for the design of improved biomass crops. The study uses an array of one- and two-dimensional solid-state NMR methods, which have collectively shown that xylan assumes primarily a three-fold screw conformation and interacts mainly with the amorphous phase of cellulose, as illustrated in Figure 6.

The study is executed with care and the paper is clearly written. The solid-state NMR experiments were done well, but involved techniques that are now fairly routinely used in the studies of plants tissues grown in a ^{13}C -enriched CO_2 atmosphere. The spectral resolution of the solid components of the cell walls was enhanced by using CP-based INADEQUATE experiment. Based on previous studies, ^{13}C resonances were assigned to various carbons in crystalline and amorphous domains of cellulose and various carbons sites in xylene and additional units attached to its mainly three-fold screw backbone. One comment and a few questions that I have here: on p. 6 the authors wrote that they also performed a DP-INADEQUATE experiment 'to detect the relatively mobile components in the secondary cell walls'. The DP experiment should in principle detect both mobile and immobile components; it is the CP process that filters away the mobile molecular species. Granted, I am not an expert on cell walls, but I would like to know what is the fraction of mobile and immobile components, what is the time scale that distinguishes the two (I assume that in the case of CP it will be motions with correlation times on the order of a ms or so) and why is this time scale so relevant here, given the weak overall nature of xylan-cellulose interactions. The authors should address these questions.

The main challenge was in the study of intermolecular contacts, as these can be only sensed through weak long-range interactions. The interactions between xylan and cellulose were inferred from the analysis of proton driven spin diffusion (PDS) spectra, again using CP as the 'mobility filter', although by its nature this technique picks up the rigid species. These experiments, in concert with T_1 relaxation measurements, elegantly showed that the interactions between xylan and cellulose involve its amorphous phase. The exact nature of these interactions is not yet known; the authors speculated that it involves both Van der Waals forces and hydrogen bonds.

One last comment: several times the authors make a generic statement that this study provides knowledge to guide future genetic engineering efforts to improve cell wall design. Please be more specific.

Overall, this work should be published, but based on solid-state NMR alone it does not rise to the standard of Nature Communications. It would have to be the gravity of new insight to wall structure that pushes it over the threshold, and I would defer this judgement to reviewers more familiar with this area of research.

Reviewer #2 (Remarks to the Author):

In this work by Gao et al. the authors show using solid-state NMR that in sorghum the highly substituted xylan adopts primarily a three-fold screw conformation which interacts with amorphous cellulose. They thus evidence a different arrangement from other plant secondary cell walls previously studied by solid-state NMR where xylan was shown to adopt primarily a two-fold screw conformation.

This work thus adds an interesting case to the information which is being gathered by the community on secondary cell walls. While the approach and methods are the same as used by the other groups working in the field, the experiments are perfectly appropriate, well designed and the results well supported. The article reads very well, the figures are of high quality and the methods well described. My only major concern is the absence of information on lignin in the article. I assume that all the information on lignin is present on the spectra already acquired by the authors? Kang et al (Nat. Comm. 10, 347 2019) have shown the importance of lignin in secondary plant cell walls of maize, in particular its interaction with xylan. I think that the role of lignin in sorghum should be addressed either by analyzing the data already gathered or, if not possible, at least discuss its potential role. Other minor points:

Page 5: Can the authors provide a hypothesis explaining the three domains found in amorphous and crystalline cellulose?

Pages 12-13: The C1 and C6 peaks of cellulose differ a lot between the amorphous and crystalline regions, can't the crystalline fraction be extracted from the 1D spectra which are likely to be more quantitative? X-ray data should also be able to provide an estimation of the sample crystallinity.

Reviewer #3 (Remarks to the Author):

Gao et al. investigated elusive grass cell wall architecture by using solid-state NMR (ssNMR) approaches. They produced ¹³C-enriched sorghum cell walls and successfully collected high-quality multi-dimensional ssNMR spectra to dissect the molecular configurations and interactions of major polysaccharides in intact secondary cell walls. Similar ssNMR-aided cell wall structural studies have been recently reported for other plant species including non-grass species, such as Arabidopsis (cited ref. #23 and #25) and softwood (ref. #24), and also several other grass species, such as maize, rice and switchgrass (ref. #25). Nevertheless, this study provides important new findings that the molecular configuration of the major xylan backbone, and their mode of interactions with cellulose chains in grass sorghum cell walls are substantially different from those previously observed in dicot and softwood cell walls. The quality of each experiment is very high and the data are overall well presented. I value this work. Having said that, I have a few comments...

Major comments:

As noted, the primary structure of hemicellulosic xylan in grass cell walls is substantially different from that in dicot cell walls. In particular, a large part of xylan backbone in grass cell walls are 2/3-O-substituted with (feruloyl)arabinosyl units, whereas dicot xylan backbone practically lacks arabinosyl substitution. However, the xylan signals assigned and characterized in this study even focusing on grass sorghum cell walls are only those from unsubstituted xylan residues. Although arabinose signals are clearly detected, signal assignments for the corresponding 2/3-O-arabinosyl substituted xylan residues are missing either in CP- and DP-INADEQUATE spectra. I think it is essential to assign them, because such arabinosyl substitution may be the key factor making the xylan configuration and xylan-cellulose interaction mode different from those in dicot, as commented by the authors and also somehow implied from an earlier work (ref. #16). I think it is not difficult, for example, by collecting GAX standard spectra, and/or predict from solution-state NMR data in literature, to assign at least substituted (etherified/glycosylated) Xn2 and/or Xn3 signals because they should be well separated from unsubstituted Xn2 and Xn3 signals. If Xn1-Xn5 in arabinosylated xylan residues could be assigned, perhaps by tracing DQ and SQ INADEQUATE cross-peaks, it would be then interesting to see their proximity to cellulose residues also in the PDS spectra.

Besides the arabinosyl substitutions on xylan, another important structural feature of grass cell walls

is the presence of ferulate units that covalently bridge xylan-xylan and xylan-lignin polymer chains. Currently, the authors comment about this only very briefly in Conclusion (P.14, Line 305). I think it should be also clearly stated in Introduction (somewhere in the second paragraph).

Minor comments:

It is interesting that the authors demonstrated that sample lyophilization diminish 3fXn and arabinose signals in CP-INADEQUATE spectra. Can the authors confirm the arabinose signals are still visible in DP-INADEQUATE spectra after lyophilization? Also add a comment about the possible change of the mobility of the arabinose residues along the conformational change of xylan backbone?

The authors used the integration of CP-PDSD cross-peaks to determine cellulose crystallinity difference between sorghum and Arabidopsis cell walls (Figure 5). Can the authors confirm this crystallinity difference also by conventional 1D CP integration (C4 or C6 peaks) and/or X-ray approaches?

P.13 Line 282. "The ratio is 0.8:1, which is consistent with..." -> "The ratio determined for Arabidopsis is 0.8:1, which is consistent with..."??

We thank all three reviewers for their positive and helpful comments. We have addressed all comments, and have explained our response to the individual points below (in blue).

Reviewer #1 (Remarks to the Author):

The study by Gao et al. explores the interactions between xylene and cellulose in sorghum, in order to better understand its wall architecture and ultimately provide guidance for the design of improved biomass crops. The study uses an array of one- and two-dimensional solid-state NMR methods, which have collectively shown that xylan assumes primarily a three-fold screw conformation and interacts mainly with the amorphous phase of cellulose, as illustrated in Figure 6.

The study is executed with care and the paper is clearly written. The solid-state NMR experiments were done well, but involved techniques that are now fairly routinely used in the studies of plants tissues grown in a ^{13}C -enriched CO_2 atmosphere. The spectral resolution of the solid components of the cell walls was enhanced by using CP-based INADEQUATE experiment. Based on previous studies, ^{13}C resonances were assigned to various carbons in crystalline and amorphous domains of cellulose and various carbons sites in xylene and additional units attached to its mainly three-fold screw backbone.

- We thank the reviewer for their careful reading of the manuscript, and for their positive review.

One comment and a few questions that I have here: on p. 6 the authors wrote that they also performed a DP-INADEQUATE experiment ‘to detect the relatively mobile components in the secondary cell walls’. The DP experiment should in principle detect both mobile and immobile components; it is the CP process that filters away the mobile molecular species.

- The target detecting cell wall components are large biopolymers, and the immobile components tend to have relatively large T_1 , $^{1-4}$ (e.g. cellulose has $T_1 \sim 5-10$ s). The DP-INADEQUATE experiments were performed with a relatively short recycle delay of 2 s, which preferentially enhanced the detection of mobile components as compared to immobile components $^{3,5-7}$. We have revised the text identified by the reviewer to clarify this.

Granted, I am not an expert on cell walls, but I would like to know what is the fraction of mobile and immobile components, what is the time scale that distinguishes the two (I assume that in the case of CP, it will be motions with correlation times on the order of a ms or so) and why is this time scale so relevant here, given the weak overall nature of xylan-cellulose interactions. The authors should address these questions.

- In the plant cell wall, cellulose is the least mobile component due to its tightly packed structure via extensive intra/intermolecular hydrogen bonding between glucan chains in

microfibrils. Published results show that the xylan molecule in plant tissues can populate both mobile and immobile states^{1,2,6,8,9}. The states can be distinguished based on the extent of their interactions with the immobile cellulose, which are mediated via either hydrogen bonding or Van der Waals forces: the cellulose-bound fraction of xylan is immobile, whereas the fraction of xylan that fills the inter-microfibril space is highly mobile^{4,6,7,9-12}. Based on our T1 measurements for carbons 2-5 of three-fold screw xylan (^{3f}Xn2-5), the average fraction of these carbons from the three-fold screw xylan with a long T1 (>2.5 s) is about 0.65. The remaining 0.35 fraction of more mobile xylan has a much shorter T1 of (<0.2 s). The differences in these T1 values indicate that 65% of the three-fold screw xylan in the sample has reduced high frequency (ns to μs timescale) molecular reorientations (side chain motions, bond vibrations, domain reorientations) compared to the other 35% of the three-fold screw xylan. The ability of these 65% of xylan molecules to cross-polarize efficiently indicates they have limited larger-scale molecular motions about an average structure with timescales on the order of μs to ms. Together with the previously published results mentioned above, the data indicate that this relatively immobile fraction of xylan is interacting extensively with the immobilized cellulose through some hydrogen bonds and van der Waals interactions. The main text was revised to clarify this. What is novel about this finding is that it is the xylan-cellulose interactions are dominated between three-fold screw xylan and amorphous cellulose in sorghum, in contrast to previous results obtained on Arabidopsis^{6,7}, maize⁷, and spruce⁹ plant cell walls, which have the similar xylan-cellulose interactions are dominated by two-fold screw xylan docking on the hydrophilic surface of crystalline cellulose with large fraction of three-fold screw xylan unassociated with the cellulose.

The main challenge was in the study of intermolecular contacts, as these can be only sensed through weak long-range interactions. The interactions between xylan and cellulose were inferred from the analysis of proton driven spin diffusion (PDS) spectra, again using CP as the 'mobility filter', although by its nature this technique picks up the rigid species. These experiments, in concert with T1 relaxation measurements, elegantly showed that the interactions between xylan and cellulose involve its amorphous phase. The exact nature of these interactions is not yet known; the authors speculated that it involves both Van der Waals forces and hydrogen bonds.

One last comment: several times the authors make a generic statement that this study provides knowledge to guide future genetic engineering efforts to improve cell wall design. Please be more specific.

- In the final paragraph of the introduction, we have now replaced the text "This will provide important insights for designing bioenergy crops that can be more efficiently deconstructed." with the following:
"Accurate cell wall models enable a predictive understanding of biomass deconstruction by identifying the most recalcitrant aspects of the architecture. Models also aid the identification of targets for engineering or breeding improved bioenergy crops. "

- In the conclusion, we have now expanded on the discussion of how the cell wall architecture we have described can support predictive design of biomass.

Overall, this work should be published, but based on solid-state NMR alone it does not rise to the standard of Nature Communications. It would have to be the gravity of new insight to wall structure that pushes it over the threshold, and I would defer this judgement to reviewers more familiar with this area of research.

Reviewer #2 (Remarks to the Author):

In this work by Gao et al. the authors show using solid-state NMR that in sorghum the highly substituted xylan adopts primarily a three-fold screw conformation which interacts with amorphous cellulose. They thus evidence a different arrangement from other plant secondary cell walls previously studied by solid-state NMR where xylan was shown to adopt primarily a two-fold screw conformation.

This work thus adds an interesting case to the information which is being gathered by the community on secondary cell walls. While the approach and methods are the same as used by the other groups working in the field, the experiments are perfectly appropriate, well designed and the results well supported. The article reads very well, the figures are of high quality and the methods well described.

- We really appreciate the reviewers' positive comments, and thank them for their time.

My only major concern is the absence of information on lignin in the article. I assume that all the information on lignin is present on the spectra already acquired by the authors? Kang et al (Nat. Comm. 10, 347 2019) have shown the importance of lignin in secondary plant cell walls of maize, in particular its interaction with xylan. I think that the role of lignin in sorghum should be addressed either by analyzing the data already gathered or, if not possible, at least discuss its potential role.

- We agree with the reviewer that lignin is an important component in the secondary cell wall. However, due to the aromatic signal suppression caused by large chemical shift anisotropy in solid state NMR⁷, and the lower lignin content of grasses (as compared to woody biomass), the sensitivity and resolution of signals from lignin is too weak to provide us any valuable information regarding lignin's role in sorghum secondary cell wall using our currently available data. In the study by Kang et al.⁷, the authors employed the dynamic nuclear polarization (DNP) technique to enhance the sensitivity and resolution of lignin signals and study the role of lignin in plant secondary cell walls. However, the plant tissue sample preparation for MAS-DNP experiments involves additional treatments to the plant tissue, such as grinding and impregnation in radical solution. We expect that this will cause irreversible changes to the native cell wall architecture, which we are trying to avoid in this effort.

- Therefore, based on current understanding of lignin in grass secondary cell walls, we first add a discussion of the other unique feature of the grass cell wall, which is the presence of high abundance of ferulic acid that can allow covalent cross-links between xylan and xylan/lignin by forming diferulates, to introduce lignin as important component that connects to matrix polysaccharides in the intact grass secondary cell walls to provide strong supporting strength to plants. Then, based on the findings of the presence of different xylan-cellulose interactions in grass cell wall compared to dicots and softwoods, we added additional discussion of the potential role of lignin in the grass secondary cell wall into the conclusion).
- Moreover, a further development of NMR experiments with enhanced sensitivity and resolution is required for understanding the role of lignin in the native plant cell walls. In addition, an engineered line of sorghum with manipulated lignin content ¹³ is being studied and compared with the wild type sorghum in our next effort to better understand the role of lignin in the native plant cell walls.

Other minor points:

Page 5: Can the authors provide a hypothesis explaining the three domains found in amorphous and crystalline cellulose?

- We thank the reviewer for this suggestion, and the following statement was added into the main results text to explain the three environments found in amorphous and crystalline cellulose.

“The three distinguished environments for both amorphous and crystalline cellulose may arise from differences in hydrogen bonding patterns between glucan chains, slight changes in glucan chain conformation, variations in bond geometries, and changes and inconsistencies in neighboring chain environments within the microfibrils ^{38,39}.”

Pages 12-13: The C1 and C6 peaks of cellulose differ a lot between the amorphous and crystalline regions, can't the crystalline fraction be extracted from the 1D spectra which are likely to be more quantitative? X-ray data should also be able to provide an estimation of the sample crystallinity.

- Although X-ray is a great tool for an estimation of crystallinity of plant cell walls, the sample preparation typically requires drying and grinding of the plant sample, which can disturb the native architecture of the wall. This is at odds with our primary objective of using solid-state NMR to characterize the native structure of the cell wall. We therefore chose to use conventional 1D ¹³C CP experiments to verify our results, as the reviewer also recommended.
- We think the reviewer meant the C4 and C6 peaks of cellulose differ a lot between the amorphous and crystalline regions, since the chemical shifts of C4 and C6 is ~90-88

ppm and ~66-64 ppm for crystalline cellulose and ~85-83 ppm and ~63-61 ppm. According to the reviewer's suggestion, we performed conventional 1D ^{13}C CP experiments on the same stem tissue samples from both sorghum and Arabidopsis and used the 1D ^{13}C CP spectra for better quantification and to verify the crystalline and amorphous cellulose ratios we previously obtained from CP-PDSD spectra.

- In 1D experiments, cellulose peaks overlap with peaks from other cell wall components, so deconvolution of the spectra is required to more accurately quantify the amount of crystalline and amorphous cellulose. Since there are fewer overlapping peaks at the chemical shift region of cellulose C4 (90-80 ppm) as compared to C6 (66-63 ppm), 1D ^{13}C CP spectra of both sorghum and Arabidopsis were deconvoluted and integrated in the C4 region to obtain the quantity of crystalline cellulose and amorphous cellulose (Figure S9). For sorghum, one major peak at 89 ppm was deconvoluted and integrated to represent the total crystalline cellulose, and the sum of the integrations of two deconvoluted peaks at 84.7 ppm and 83.8 ppm represent the total quantity of amorphous cellulose. For Arabidopsis, the sum of integrations of two deconvoluted peaks at 89.2 ppm and 88.2 ppm to represent the total crystalline cellulose, and the sum of the integrations of two deconvoluted peaks at 84.7 ppm and 83.8 ppm represent the total amorphous cellulose. The crystalline to amorphous cellulose ratio obtained was 1:3.1 for sorghum and 0.73:1 for Arabidopsis, which is consistent with our previous result

obtained from the integrations of CP-PDSD cross peaks (C1-C4). A new Figure S9 (see below) was added into the Supplementary Materials, and also the main results text was revised to describe these additional 1D ^{13}C CP data.

New Figure S9. 1D ^{13}C CP experiments on the stem tissue of sorghum and Arabidopsis (spectra were zoomed at cellulose C4 region, 90-80 ppm). Spectra were deconvoluted and integrated by a built-in Global Spectral

Deconvolution (GSD) method in MestReNova NMR processing software (Version 14.1.0). The crystalline to amorphous cellulose ratio is determined as 1:3.1 for sorghum and 0.73:1 for Arabidopsis, which is consistent with CP-PDSD determined results.

Reviewer #3 (Remarks to the Author):

Gao et al. investigated elusive grass cell wall architecture by using solid-state NMR (ssNMR) approaches. They produced ^{13}C -enriched sorghum cell walls and successfully collected high-quality multi-dimensional ssNMR spectra to dissect the molecular configurations and interactions of major polysaccharides in intact secondary cell walls. Similar ssNMR-aided cell wall structural studies have been recently reported for other plant species including non-grass species, such as *Arabidopsis* (cited ref. #23 and #25) and softwood (ref. #24), and also several other grass species, such as maize, rice and switchgrass (ref. #25). Nevertheless, this study provides important new findings that the molecular configuration of the major xylan backbone, and their mode of interactions with cellulose chains in grass sorghum cell walls are substantially different from those previously observed in dicot and softwood cell walls. The quality of each experiment is very high and the data are overall well presented. I value this work. Having said that, I have a few comments...

Major comments:

As noted, the primary structure of hemicellulosic xylan in grass cell walls is substantially different from that in dicot cell walls. In particular, a large part of xylan backbone in grass cell walls are 2/3-O-substituted with (feruloyl)arabinosyl units, whereas dicot xylan backbone practically lacks arabinosyl substitution. However, the xylan signals assigned and characterized in this study even focusing on grass sorghum cell walls are only those from unsubstituted xylan residues. Although arabinose signals are clearly detected, signal assignments for the corresponding 2/3-O-arabinosyl substituted xylan residues are missing either in CP- and DP-INADEQUATE spectra. I think it is essential to assign them, because such arabinosyl substitution may be the key factor making the xylan configuration and xylan-cellulose interaction mode different from those in dicot, as commented by the authors and also somehow implied from an earlier work (ref. #16). I think it is not difficult, for example, by collecting GAX standard spectra, and/or predict from solution-state NMR data in literature, to assign at least substituted (etherified/glycosylated) Xn2 and/or Xn3 signals because they should be well separated from unsubstituted Xn2 and Xn3 signals. If Xn1-Xn5 in arabinosylated xylan residues could be assigned, perhaps by tracing DQ and SQ INADEQUATE cross-peaks, it would be then interesting to see their proximity to cellulose residues also in the PDS spectra.

- We thank the reviewer for this excellent point. In grass cell walls, arabinosyl substitutions on xylan backbone are dominated by the 3-O linkages, and the 2-O-arabinosyl substitution mostly occur when xylosyl units from xylan backbone were doubly substituted with arabinosyl units at both O-2 and O-3 position ¹⁴⁻²⁰. It should be noted that while these double substitutions are abundant in soluble xylans, such as wheat flour arabinoxylan, they are much less frequent in cell wall GAX. Hence, the 2-O-arabinose substituted xylosyl unit is present in much lower abundance than the 3-O-arabinose substituted xylosyl unit. This is also reflected in our NMR result as well.

- Signals from the 3-O-arabinose substituted xylosyl units of xylan were assigned in our both CP- and DP-INADEQUATE spectra of the sorghum stem internodes according to the reference of chemical shift assignments published by Wang et al.'s study on *Brachypodium* primary cell wall with solid-state NMR³ and by Komatsu et al.'s²¹ study on the corn cell wall with solution-state NMR. Chemical shifts of the five carbons of 3-O-arabinose substituted xylosyl units were added to Table S2 and labelled in Figure 1 in the main text and Figures S3 in the Supplementary Materials as ^{3f,A}Xn1 to ^{3f,A}Xn5 (revised Figures are shown below). However, due to the low abundance, no signals from 2-O-arabinose substituted xylosyl units were detected by our INADEQUATE experiments. Additionally, we believe these 3-O-arabinose substituted xylosyl units of xylan are mainly present in three-fold screw conformation of xylan, because the chemical shifts of ^{3f,A}Xn1 and ^{3f,A}Xn4 from 3-O-arabinose substituted xylosyl units remain similar to the ^{3f}Xn1 and ^{3f}Xn4 from unsubstituted xylosyl units of three-fold screw xylan⁶. This supports our conclusion that extensive arabinosyl substitutions on xylan backbone impede the formation of two-fold screw xylan. The main text was revised in the results to describe the new assignments of 3-O-arabinose substituted xylosyl units. The Figure 1 caption was revised to describe the labeling of ^{3f,A}Xn1-5 in Figure 1.

Edited Figure 1. Immobile polysaccharides detected by refocused ¹³C CP-INADEQUATE experiments in sorghum secondary cell walls. The chemical shift assignments shown are derived from previously published chemical shifts (listed in Supplementary Table S2). Signals

from pectin and xyloglucan are not detected due to their relatively high mobility, as well as their low abundance in the grass secondary cell wall. Glucosyl units (**C1-C6**) from cellulose in six distinguishable environments were assigned into two major domains, which are the crystalline ^{13}C and amorphous ^{13}C cellulose (^{1-3}C **C1**- ^{1-3}C **C6**, $^{1-3\text{A}}$ **C1**- $^{1-3\text{A}}$ **C6**). A lack of signals from xylosyl units of two-fold screw xylan was observed. Instead, signals from immobile xylosyl units of rigid three-fold screw xylan were detected ($^{3\text{f}}$ **Xn1**- $^{3\text{f}}$ **Xn5**, $^{3\text{f,A}}$ **Xn1**- $^{3\text{f,A}}$ **Xn5**). High intensity signals from xylan arabinosyl units were also detected (**A1-A5**).

Edited Figure S3. Mobile polysaccharides detected by refocused ^{13}C DP-INADEQUATE experiments with recycle delay of 2 s in sorghum secondary cell walls.

- Moreover, according to the new assigned chemical shifts of 3-O-arabinose substituted xylosyl units, the chemical shift of $^{3\text{f,A}}$ **Xn3** (76.7 ppm) is significantly distinct from the all the chemical shifts from the five carbons of unsubstituted xylosyl units in three-fold screw xylan. This observation can then be used to probe the interaction between xylan and cellulose via the 3-O-arabinose substituted xylosyl units. Indeed, in the CP-PDSD spectrum with short mixing time, 30 ms (Figure 2), additional cross peaks are identified, such as $^{3\text{f,A}}$ **Xn3**-**C1** (76.7 ppm, 105.2 ppm), $^{3\text{f,A}}$ **Xn3**- $^{2\text{A}}$ **C4** (76.7 ppm, 84.1 ppm), and $^{3\text{f,A}}$ **Xn3**- $^{3\text{A}}$ **C6** (76.7 ppm, 62.5 ppm), which indicate additional close interactions between three-fold screw xylan and amorphous cellulose through the 3-O-arabinose substituted xylosyl units. Consistent with our other data, there are no cross peaks detected that

indicate the interactions between three-fold screw xylan and crystalline cellulose. This further supports our results that there is a close interaction between three-fold screw xylan and amorphous cellulose. Similar cross peaks are also identified in the CP-PDSD spectrum with a longer mixing time (Figure S7). Figure 2 in the main text and Figure S7 in the Supplementary Materials were revised with the newly identified cross peaks (shown below). The main text was revised in the results to describe the cross peaks identified between 3-O-arabinose substituted xylosyl units of three-fold screw xylan and amorphous cellulose.

Edited Figure 2. Close proximities between moieties from three-fold screw xylan and amorphous cellulose were detected in sorghum secondary cell walls by CP-PDSD experiments with a short mixing time (30 ms). Cross peaks indicate that the majority of intramolecular interactions and close intermolecular interactions between arabinosyl and xylosyl units from three-fold screw xylan and glucosyl units from amorphous cellulose were labelled in the spectrum.

Edited Figure S7. Close proximities between moieties from three-fold screw xylan and amorphous cellulose were identified in sorghum secondary cell walls by CP-PDSD experiments with a mixing time of 100 ms. Similar to the CP-PDSD spectrum with short mixing time (30 ms), cross peaks that indicate interactions between arabinosyl and xylosyl units from three-fold screw xylan and glucosyl units from amorphous cellulose were detected with enhanced intensities and labelled in the spectrum.

Besides the arabinosyl substitutions on xylan, another important structural feature of grass cell walls is the presence of ferulate units that covalently bridge xylan-xylan and xylan-lignin polymer chains. Currently, the authors comment about this only very briefly in Conclusion (P.14, Line 305). I think it should be also clearly stated in Introduction (somewhere in the second paragraph).

- We thank the reviewer for this good point. We have now included a discussion of the presence and role of ferulate units in grass cell walls to the second paragraph of the introduction as suggested,
- “Additionally, another characteristic feature of grass secondary cell wall is the presence of hydroxycinnamic acids (i.e., ferulic acid and *p*-coumaric acid), which are esterified to the O-5 of the α -1,3 arabinosyl substitutions on grass xylan backbone and can be esterified and/or etherified to lignin^{10,11,13,17,18}. Ferulate units on grass xylan facilitate the covalent cross-link of xylan-xylan and xylan-lignin by formation of diferulates through

dehydrodimerization and participation in lignification via radical coupling to bridge the lignin and polysaccharides and stiffen the cell wall and maintain structural integrity^{11,18-23} .

Minor comments:

It is interesting that the authors demonstrated that sample lyophilization diminish 3fXn and arabinose signals in CP-INADEQUATE spectra. Can the authors confirm the arabinose signals are still visible in DP-INADEQUATE spectra after lyophilization? Also add a comment about the possible change of the mobility of the arabinose residues along the conformational change of xylan backbone?

- We thank the reviewer for this comment. As the reviewer suggested, we also performed the DP-INADEQUATE experiments on both untreated native and lyophilized-rehydrated stem tissue samples. Below is new Figure S5, which shows a spectral overlay of refocused ¹³C DP-INADEQUATE experiments on native untreated and lyophilized-rehydrated sorghum stem tissue, which is now included in the Supplementary Materials. Not only are the arabinose signals still visible in DP-INADEQUATE, but the spectra shows that there is a significant enhancement of arabinose and xylose signals from the three-fold screw xylan after lyophilization. This indicates that a significant amount of arabinosyl and xylosyl units from three-fold screw xylan have become more mobile after lyophilization and rehydration of the sample. This verifies our hypothesis that lyophilization and rehydration may free the relatively immobile three-fold screw xylan from interactions with other components in the native cell wall, increasing mobility, and enhancing its detection in DP-INADEQUATE experiments. The main text was revised at to clarify this.

Figure S5. Spectral overlay of refocused ^{13}C DP-INADEQUATE experiments on native untreated (black) and lyophilized-rehydrated (yellow) sorghum stem tissue. In contrast to the ^{13}C CP-INADEQUATE experiments (Figure S4), the intensities of arabinose signals (i.e., A1-A5) and xylose signals from three-fold screw xylan (e.g., $^{3f}\text{Xn4}$ and $^{3f}\text{Xn5}$, $^{3f,A}\text{Xn4}$ and $^{3f,A}\text{Xn5}$) in the lyophilized-rehydrated sample were significantly increased compared to the untreated sample.

The authors used the integration of CP-PDSD cross-peaks to determine cellulose crystallinity difference between sorghum and Arabidopsis cell walls (Figure 5). Can the authors confirm this crystallinity difference also by conventional 1D CP integration (C4 or C6 peaks) and/or X-ray approaches?

- We thank the reviewer for this comment. This question is very similar to the point made by reviewer #2, as described above.

P.13 Line 282. “The ratio is 0.8:1, which is consistent with...” -> “The ratio determined for Arabidopsis is 0.8:1, which is consistent with...”??

- We thank the reviewer for pointing this out. The main text was revised according to the reviewer's suggestion.

References

1. Wang, T., Phyto, P. & Hong, M. Multidimensional solid-state NMR spectroscopy of plant cell walls. *Solid State Nucl. Magn. Reson.* **78**, 56–63 (2016).
2. Dick-Pérez, M. *et al.* Structure and interactions of plant cell-wall polysaccharides by two- and three-dimensional magic-angle-spinning solid-state NMR. *Biochemistry* **50**, 989–1000 (2011).
3. Wang, T., Salazar, A., Zobotina, O. A. & Hong, M. Structure and dynamics of Brachypodium primary cell wall polysaccharides from two-dimensional ^{13}C solid-state nuclear magnetic resonance spectroscopy. *Biochemistry* **53**, 2840–2854 (2014).
4. Wang, T., Park, Y. B., Cosgrove, D. J. & Hong, M. Cellulose-Pectin Spatial Contacts Are Inherent to Never-Dried 19 Arabidopsis thaliana Primary Cell Walls: Evidence from Solid-State ^{13}C NMR. (2015).
5. Dupree, R. *et al.* Probing the molecular architecture of Arabidopsis thaliana secondary cell walls using two- and three-dimensional ^{13}C solid state nuclear magnetic resonance spectroscopy. *Biochemistry* **54**, 2335–2345 (2015).
6. Simmons, T. J. *et al.* Folding of xylan onto cellulose fibrils in plant cell walls revealed by solid-state NMR. *Nat. Commun.* **7**, 13902 (2016).
7. Kang, X. *et al.* Lignin-polysaccharide interactions in plant secondary cell walls revealed by solid-state NMR. *Nat. Commun.* **10**, 347 (2019).
8. Wang, T. & Hong, M. Solid-state NMR investigations of cellulose structure and interactions with matrix polysaccharides in plant primary cell walls. *J. Exp. Bot.* **67**, 503–514 (2016).
9. Terrett, O. M. *et al.* Molecular architecture of softwood revealed by solid-state NMR. *Nat. Commun.* **10**, 4978 (2019).
10. Phyto, P., Wang, T., Xiao, C., Anderson, C. T. & Hong, M. Effects of Pectin Molecular Weight Changes on the Structure, Dynamics, and Polysaccharide Interactions of Primary Cell Walls of Arabidopsis thaliana: Insights from Solid-State NMR. *Biomacromolecules* **18**, 2937–2950 (2017).
11. Wang, T. & Hong, M. in *NMR in Glycoscience and Glycotechnology* 290–304 (2017).
12. Wang, T., Williams, J. K., Schmidt-Rohr, K. & Hong, M. Relaxation-compensated difference spin diffusion NMR for detecting ^{13}C - ^{13}C long-range correlations in proteins and polysaccharides. *J. Biomol. NMR* **61**, 97–107 (2015).
13. Eudes, A. *et al.* Expression of a bacterial 3-dehydroshikimate dehydratase reduces lignin content and improves biomass saccharification efficiency. *Plant Biotechnol. J.* **13**, 1241–1250 (2015).
14. Ebringerová, A., Hromádková, Z., Heinze, T. & Hemicellulose, T. H. Polysaccharides I. *Adv Polym Sci* **186**, 67 (2005).
15. Scheller, H. V. & Ulvskov, P. Hemicelluloses. *Annu. Rev. Plant Biol.* **61**, 263–289 (2010).
16. Höjje, A. *et al.* Evidence of the presence of 2-O- β -d-xylopyranosyl- α -l-arabinofuranose side chains in barley husk arabinoxylan. *Carbohydr. Res.* **341**, 2959–2966 (2006).
17. Wende, G. & Fry, S. C. 2-O- β -d-xylopyranosyl-(5-O-feruloyl)-l-arabinose, a widespread component of grass cell walls. *Phytochemistry* **44**, 1019–1030 (1997).
18. Verbruggen, M. A. *Glucuronoarabinoxylans from sorghum grain*. (Verbruggen, 1996).

19. Nilsson, M. Water unextractable polysaccharides from three milling fractions of rye grain. *Carbohydr. Polym.* **30**, 229–237 (1996).
20. Huisman, M. M. H., Schols, H. A. & Voragen, A. G. J. Glucuronoarabinoxylans from maize kernel cell walls are more complex than those from sorghum kernel cell walls. *Carbohydr. Polym.* **43**, 269–279 (2000).
21. Komatsu, T. & Kikuchi, J. Comprehensive signal assignment of ¹³C-labeled lignocellulose using multidimensional solution NMR and ¹³C chemical shift comparison with solid-state NMR. *Anal. Chem.* **85**, 8857–8865 (2013).

REVIEWERS' COMMENTS

Reviewer #1 (Remarks to the Author):

My concerns expressed in the first review were fully addressed by the authors. I do not have any additional comments.

Reviewer #2 (Remarks to the Author):

All my comments have been addressed. I have read the article with the modifications which address all reviewers comments and think the manuscript is now ready for publication.

Reviewer #3 (Remarks to the Author):

I thank the authors' efforts for addressing all the reviewers' comments in such details. With respect to my comments on the previous version, I particularly appreciate the new assignment and analysis of the 3-O-arabinose-substituted xylan residues, which I believe is an additional key finding of this study focusing on the analysis of grass cell walls. In addition, the additional analysis of the rearrangement of arabinosyl and xylosyl units from three-fold screw xylan over the lyophilization/rehydration process further strengthened the manuscript. Given these significant improvements, I am supportive of the publication of the current version in the journal. Yuki Tobimatsu